# Online Learning of Dynamic Parameters in Social Networks

**Shahin Shahrampour** [1]     **Alexander Rakhlin** [2]     **Ali Jadbabaie** [1]

[1]Department of Electrical and Systems Engineering, [2]Department of Statistics
University of Pennsylvania
Philadelphia, PA 19104 USA
[1]{shahin,jadbabai}@seas.upenn.edu  [2]rakhlin@wharton.upenn.edu

## Abstract

This paper addresses the problem of online learning in a dynamic setting. We consider a social network in which each individual observes a private signal about the underlying state of the world and communicates with her neighbors at each time period. Unlike many existing approaches, the underlying state is dynamic, and evolves according to a geometric random walk. We view the scenario as an optimization problem where agents aim to learn the true state while suffering the smallest possible loss. Based on the decomposition of the global loss function, we introduce two update mechanisms, each of which generates an estimate of the true state. We establish a tight bound on the rate of change of the underlying state, under which individuals can track the parameter with a bounded variance. Then, we characterize explicit expressions for the steady state mean-square deviation(MSD) of the estimates from the truth, per individual. We observe that only one of the estimators recovers the optimal MSD, which underscores the impact of the objective function decomposition on the learning quality. Finally, we provide an upper bound on the regret of the proposed methods, measured as an average of errors in estimating the parameter in a finite time.

## 1 Introduction

In recent years, distributed estimation, learning and prediction has attracted a considerable attention in wide variety of disciplines with applications ranging from sensor networks to social and economic networks [1–6]. In this broad class of problems, agents aim to learn the true value of a parameter often called the *underlying state of the world*. The state could represent a product, an opinion, a vote, or a quantity of interest in a sensor network. Each agent observes a *private* signal about the underlying state at each time period, and communicates with her neighbors to augment her imperfect observations. Despite the wealth of research in this area when the underlying state is fixed (see e.g. [1–3, 7]), often the state is subject to some change over time(e.g. the price of stocks) [8–11]. Therefore, it is more realistic to study models which allow the parameter of interest to vary. In the non-distributed context, such models have been studied in the classical literature on time-series prediction, and, more recently, in the literature on online learning under relaxed assumptions about the nature of sequences [12]. In this paper we aim to study the sequential prediction problem in the context of a social network and noisy feedback to agents.

We consider a stochastic optimization framework to describe an online social learning problem when the underlying state of the world varies over time. Our motivation for the current study is the results of [8] and [9] where authors propose a social learning scheme in which the underlying state follows a simple random walk. However, unlike [8] and [9], we assume a *geometric* random walk evolution with an associated *rate of change*. This enables us to investigate the interplay of social learning, network structure, and the rate of state change, especially in the interesting case that the rate is

greater than unity. We then pose the social learning as an optimization problem in which individuals aim to suffer the smallest possible loss as they observe the stream of signals. Of particular relevance to this work is the work of Duchi *et al.* in [13] where the authors develop a distributed method based on dual averaging of sub-gradients to converge to the optimal solution. In this paper, we restrict our attention to quadratic loss functions regularized by a quadratic proximal function, but there is no fixed optimal solution as the underlying state is dynamic. In this direction, the key observation is the decomposition of the global loss function into local loss functions. We consider two decompositions for the global objective, each of which gives rise to a *single-consensus-step* belief update mechanism. The first method incorporates the averaged prior beliefs among neighbors with the new private observation, while the second one takes into account the observations in the neighborhood as well. In both scenarios, we establish that the estimates are eventually unbiased, and we characterize an explicit expression for the mean-square deviation(MSD) of the beliefs from the truth, per individual. Interestingly, this quantity relies on the whole spectrum of the communication matrix which exhibits the formidable role of the network structure in the asymptotic learning. We observe that the estimators outperform the upper bound provided for MSD in the previous work [8]. Furthermore, only one of the two proposed estimators can compete with the centralized optimal Kalman Filter [14] in certain circumstances. This fact underscores the dependence of optimality on decomposition of the global loss function. We further highlight the influence of connectivity on learning by quantifying the ratio of MSD for a complete versus a disconnected network. We see that this ratio is always less than unity and it can get arbitrarily close to zero under some constraints.

Our next contribution is to provide an upper bound for *regret* of the proposed methods, defined as an average of errors in estimating the parameter up to a given time minus the long-run expected loss due to noise and dynamics alone. This finite-time regret analysis is based on the recently developed concentration inequalities for matrices and it complements the asymptotic statements about the behavior of MSD.

Finally, we examine the trade-off between the network sparsity and learning quality in a microscopic level. Under mild technical constraints, we see that losing each connection has detrimental effect on learning as it monotonically increases the MSD. On the other hand, capturing agents communications with a graph, we introduce the notion of *optimal edge* as the edge whose addition has the most effect on learning in the sense of MSD reduction. We prove that such a friendship is likely to occur between a pair of individuals with high self-reliance that have the least common neighbors.

## 2 Preliminaries

### 2.1 State and Observation Model

We consider a network consisting of a finite number of agents $\mathcal{V} = \{1, 2, ..., N\}$. The agents indexed by $i \in \mathcal{V}$ seek *the underlying state of the world*, $x_t \in \mathbb{R}$, which varies over time and evolves according to

$$x_{t+1} = ax_t + r_t, \tag{1}$$

where $r_t$ is a zero mean *innovation*, which is independent over time with finite variance $\mathbb{E}[r_t^2] = \sigma_r^2$, and $a \in \mathbb{R}$ is the expected *rate of change* of the state of the world, assumed to be available to all agents, and could potentially be greater than unity. We assume the *initial state* $x_0$ is a finite random variable drawn independently by the nature. At time period $t$, each agent $i$ receives a private signal $y_{i,t} \in \mathbb{R}$, which is a noisy version of $x_t$, and can be described by the linear equation

$$y_{i,t} = x_t + w_{i,t}, \tag{2}$$

where $w_{i,t}$ is a zero mean *observation noise* with finite variance $\mathbb{E}[w_{i,t}^2] = \sigma_w^2$, and it is assumed to be independent over time and agents, and uncorrelated to the innovation noise. Each agent $i$ forms an *estimate* or a *belief* about the true value of $x_t$ at time $t$ conforming to an update mechanism that will be discussed later. Much of the difficulty of this problem stems from the hardness of tracking a dynamic state with noisy observations, especially when $|a| > 1$, and communication mitigates the difficulty by virtue of reducing the effective noise.

## 2.2 Communication Structure

Agents communicate with each other to update their beliefs about the underlying state of the world. The interaction between agents is captured by an undirected graph $\mathcal{G} = (\mathcal{V}, \mathcal{E})$, where $\mathcal{V}$ is the set of agents, and if there is a link between agent $i$ and agent $j$, then $\{i, j\} \in \mathcal{E}$. We let $\bar{\mathcal{N}}_i = \{j \in \mathcal{V} : \{i, j\} \in \mathcal{E}\}$ be the set of neighbors of agent $i$, and $\mathcal{N}_i = \bar{\mathcal{N}}_i \cup \{i\}$. Each agent $i$ can only communicate with her neighbors, and assigns a weight $p_{ij} > 0$ for any $j \in \bar{\mathcal{N}}_i$. We also let $p_{ii} \geq 0$ denote the *self-reliance* of agent $i$.

**Assumption 1.** *The communication matrix $P = [p_{ij}]$ is symmetric and doubly stochastic, i.e., it satisfies*

$$p_{ij} \geq 0 \quad , \quad p_{ij} = p_{ji} \quad , \quad and \quad \sum_{j \in \mathcal{N}_i} p_{ij} = \sum_{j=1}^{N} p_{ij} = 1.$$

*We further assume the eigenvalues of $P$ are in descending order and satisfy*

$$-1 < \lambda_N(P) \leq ... \leq \lambda_2(P) < \lambda_1(P) = 1.$$

## 2.3 Estimate Updates

The goal of agents is to learn $x_t$ in a collaborative manner by making sequential predictions. From optimization perspective, this can be cast as a quest for online minimization of the separable, global, time-varying cost function

$$\min_{\bar{x} \in \mathbb{R}} f_t(\bar{x}) = \frac{1}{N} \sum_{i=1}^{N} \left( \hat{f}_{i,t}(\bar{x}) \triangleq \frac{1}{2} \mathbb{E}(y_{i,t} - \bar{x})^2 \right) = \frac{1}{N} \sum_{i=1}^{N} \left( \tilde{f}_{i,t}(\bar{x}) \triangleq \sum_{j=1}^{N} p_{ij} \hat{f}_{j,t}(\bar{x}) \right), \quad (3)$$

at each time period $t$. One approach to tackle the stochastic learning problem formulated above is to employ *distributed dual averaging* regularized by a quadratic *proximal function* [13]. To this end, if agent $i$ exploits $\hat{f}_{i,t}$ as the local loss function, she updates her belief as

$$\hat{x}_{i,t+1} = a\bigg( \underbrace{\sum_{j \in \mathcal{N}_i} p_{ij} \hat{x}_{j,t}}_{\text{consensus update}} + \underbrace{\alpha(y_{i,t} - \hat{x}_{i,t})}_{\text{innovation update}} \bigg), \quad (4)$$

while using $\tilde{f}_{i,t}$ as the local loss function results in the following update

$$\tilde{x}_{i,t+1} = a\bigg( \underbrace{\sum_{j \in \mathcal{N}_i} p_{ij} \tilde{x}_{j,t}}_{\text{consensus update}} + \underbrace{\alpha(\sum_{j \in \mathcal{N}_i} p_{ij} y_{j,t} - \tilde{x}_{i,t})}_{\text{innovation update}} \bigg), \quad (5)$$

where $\alpha \in (0, 1]$ is a constant step size that agents place for their innovation update, and we refer to it as *signal weight*. Equations (4) and (5) are distinct, *single-consensus-step* estimators differing in the choice of the local loss function with (4) using only private observations while (5) averaging observations over the neighborhood. We analyze both class of estimators noting that one might expect (5) to perform better than (4) due to more information availability.

Note that the choice of constant step size provides an insight on the interplay of persistent innovation and learning abilities of the network. We remark that agents can easily *learn* the fixed rate of change $a$ by taking ratios of observations, and we assume that this has been already performed by the agents in the past. The case of a changing $a$ is beyond the scope of the present paper. We also point out that the real-valued (rather than vector-valued) nature of the state is a simplification that forms a clean playground for the study of the effects of social learning, effects of friendships, and other properties of the problem.

## 2.4 Error Process

Defining the local error processes $\hat{\xi}_{i,t}$ and $\tilde{\xi}_{i,t}$, at time $t$ for agent $i$, as

$$\hat{\xi}_{i,t} \triangleq \hat{x}_{i,t} - x_t \quad \text{and} \quad \tilde{\xi}_{i,t} \triangleq \tilde{x}_{i,t} - x_t,$$

and stacking the local errors in vectors $\hat{\xi}_t, \tilde{\xi}_t \in \mathbb{R}^N$, respectively, such that

$$\hat{\xi}_t \triangleq [\hat{\xi}_{1,t}, ..., \hat{\xi}_{N,t}]^\mathsf{T} \quad \text{and} \quad \tilde{\xi}_t \triangleq [\tilde{\xi}_{1,t}, ..., \tilde{\xi}_{N,t}]^\mathsf{T}, \tag{6}$$

one can show that the aforementioned collective error processes could be described as a linear dynamical system.

**Lemma 2.** *Given Assumption 1, the collective error processes $\hat{\xi}_t$ and $\tilde{\xi}_t$ defined in* (6) *satisfy*

$$\hat{\xi}_{t+1} = Q\hat{\xi}_t + \hat{s}_t \quad \text{and} \quad \tilde{\xi}_{t+1} = Q\tilde{\xi}_t + \tilde{s}_t, \tag{7}$$

*respectively, where*

$$Q = a(P - \alpha I_N), \tag{8}$$

*and*

$$\hat{s}_t = (\alpha a)[w_{1,t}, ..., w_{N,t}]^\mathsf{T} - r_t \mathbf{1}_N \quad \text{and} \quad \tilde{s}_t = (\alpha a)P[w_{1,t}, ..., w_{N,t}]^\mathsf{T} - r_t \mathbf{1}_N, \tag{9}$$

*with $\mathbf{1}_N$ being vector of all ones.*

Throughout the paper, we let $\rho(Q)$, denote the spectral radius of $Q$, which is equal to the largest singular value of $Q$ due to symmetry.

# 3 Social Learning: Convergence of Beliefs and Regret Analysis

In this section, we study the behavior of estimators (4) and (5) in the mean and mean-square sense, and we provide the regret analysis.

In the following proposition, we establish a tight bound for $a$, under which agents can achieve asymptotically unbiased estimates using proper signal weight.

**Proposition 3** (Unbiased Estimates)**.** *Given the network $\mathcal{G}$ with corresponding communication matrix $P$ satisfying Assumption 1, the rate of change of the social network in* (4) *and* (5) *must respect the constraint*

$$|a| < \frac{2}{1 - \lambda_N(P)},$$

*to allow agents to form asymptotically unbiased estimates of the underlying state.*

Proposition 3 determines the trade-off between the rate of change and the network structure. In other words, changing less than the rate given in the statement of the proposition, individuals can always track $x_t$ with bounded variance by selecting an appropriate signal weight. However, the proposition does not make any statement on the learning quality. To capture that, we define the steady state Mean Square Deviation(MSD) of the network from the truth as follows.

**Definition 4** ((Steady State-)Mean Square Deviation)**.** *Given the network $\mathcal{G}$ with a rate of change which allows unbiased estimation, the steady state of the error processes in* (7) *is defined as follows*

$$\hat{\Sigma} \triangleq \lim_{t\to\infty} \mathbb{E}[\hat{\xi}_t \hat{\xi}_t^\mathsf{T}] \quad \text{and} \quad \tilde{\Sigma} \triangleq \lim_{t\to\infty} \mathbb{E}[\tilde{\xi}_t \tilde{\xi}_t^\mathsf{T}].$$

*Hence, the (Steady State-)Mean Square Deviation of the network is the deviation from the truth in the mean-square sense, per individual, and it is defined as*

$$\hat{MSD} \triangleq \frac{1}{N} Tr(\hat{\Sigma}) \quad \text{and} \quad \tilde{MSD} \triangleq \frac{1}{N} Tr(\tilde{\Sigma}).$$

**Theorem 5** (MSD)**.** *Given the error processes* (7) *with $\rho(Q) < 1$, the steady state MSD for* (4) *and* (5) *is a function of the communication matrix $P$, and the signal weight $\alpha$ as follows*

$$\hat{MSD}(P, \alpha) = R_{MSD}(\alpha) + \hat{W}_{MSD}(P, \alpha) \qquad \tilde{MSD}(P, \alpha) = R_{MSD}(\alpha) + \tilde{W}_{MSD}(P, \alpha), \tag{10}$$

*where*

$$R_{MSD}(\alpha) \triangleq \frac{\sigma_r^2}{1 - a^2(1 - \alpha)^2}, \tag{11}$$

*and*

$$\hat{W}_{MSD}(P, \alpha) \triangleq \frac{1}{N} \sum_{i=1}^{N} \frac{a^2 \alpha^2 \sigma_w^2}{1 - a^2(\lambda_i(P) - \alpha)^2} \quad \text{and} \quad \tilde{W}_{MSD}(P, \alpha) \triangleq \frac{1}{N} \sum_{i=1}^{N} \frac{a^2 \alpha^2 \sigma_w^2 \lambda_i^2(P)}{1 - a^2(\lambda_i(P) - \alpha)^2}. \tag{12}$$

Theorem 5 shows that the steady state MSD is governed by all eigenvalues of $P$ contributing to $W_{MSD}$ pertaining to the observation noise, while $R_{MSD}$ is the penalty incurred due to the innovation noise. Moreover, (5) outperforms (4) due to richer information diffusion, which stresses the importance of global loss function decomposition.

One might advance a conjecture that a complete network, where all individuals can communicate with each other, achieves a lower steady state MSD in the learning process since it provides the most information diffusion among other networks. This intuitive idea is discussed in the following corollary beside a few examples.

**Corollary 6.** *Denoting the complete, star, and cycle graphs on $N$ vertices by $K_N$, $S_N$, and $C_N$, respectively, and denoting their corresponding Laplacians by $L_{K_N}$, $L_{S_N}$, and $L_{C_N}$, under conditions of Theorem 5,*

*(a) For $P = I - \frac{1-\alpha}{N} L_{K_N}$, we have*

$$\lim_{N\to\infty} \hat{MSD}_{K_N} = R_{MSD}(\alpha) + a^2\alpha^2\sigma_w^2. \tag{13}$$

*(b) For $P = I - \frac{1-\alpha}{N} L_{S_N}$, we have*

$$\lim_{N\to\infty} \hat{MSD}_{S_N} = R_{MSD}(\alpha) + \frac{a^2\alpha^2\sigma_w^2}{1 - a^2(1-\alpha)^2}. \tag{14}$$

*(c) For $P = I - \beta L_{C_N}$, where $\beta$ must preserve unbiasedness, we have*

$$\lim_{N\to\infty} \hat{MSD}_{C_N} = R_{MSD}(\alpha) + \int_0^{2\pi} \frac{a^2\alpha^2\sigma_w^2}{1 - a^2(1 - \beta(2 - 2\cos(\tau)) - \alpha)^2} \frac{d_\tau}{2\pi}. \tag{15}$$

*(d) For $P = I - \frac{1}{N} L_{K_N}$, we have*

$$\lim_{N\to\infty} \tilde{MSD}_{K_N} = R_{MSD}(\alpha). \tag{16}$$

*Proof.* Noting that the spectrum of $L_{K_N}$, $L_{S_N}$ and $L_{C_N}$ are, respectively [15], $\{\lambda_N = 0, \lambda_{N-1} = N, ..., \lambda_1 = N\}$, $\{\lambda_N = 0, \lambda_{N-1} = 1, ..., \lambda_2 = 1, \lambda_1 = N\}$, and $\{\lambda_i = 2 - 2\cos(\frac{2\pi i}{N})\}_{i=0}^{N-1}$, substituting each case in (10), and taking the limit over $N$, the proof follows immediately. □

To study the effect of communication let us consider the estimator (4). Under purview of Theorem 5 and Corollary 6, the ratio of the steady state MSD for a complete network (13) versus a fully disconnected network($P = I_N$) can be computed as

$$\lim_{N\to\infty} \frac{\hat{MSD}_{K_N}}{\hat{MSD}_{disconnected}} = \frac{\sigma_r^2 + a^2\alpha^2\sigma_w^2(1 - a^2(1-\alpha)^2)}{\sigma_r^2 + a^2\alpha^2\sigma_w^2} \approx 1 - a^2(1-\alpha)^2,$$

for $\sigma_r^2 \ll \sigma_w^2$. The ratio above can get arbitrary close to zero which, indeed, highlights the influence of communication on the learning quality.

We now consider Kalman Filter(KF) [14] as the optimal centralized counterpart of (5). It is well-known that the steady state KF satisfies a Riccati equation, and when the parameter of interest is scalar, the Riccati equation simplifies to a quadratic with the positive root

$$\Sigma_{KF} = \frac{a^2\sigma_w^2 - \sigma_w^2 + N\sigma_r^2 + \sqrt{(a^2\sigma_w^2 - \sigma_w^2 + N\sigma_r^2)^2 + 4N\sigma_w^2\sigma_r^2}}{2N}.$$

Therefore, comparing with the complete graph (16), we have

$$\lim_{N\to\infty} \Sigma_{KF} = \sigma_r^2 \le \frac{\sigma_r^2}{1 - a^2(1-\alpha)^2},$$

and the upper bound can be made tight by choosing $\alpha = 1$ for $|a| < \frac{1}{|\lambda_N(P)-1|}$. If $|a| \ge \frac{1}{|\lambda_N(P)-1|}$ we should choose an $\alpha < 1$ to preserve unbiasedness as well.

On the other hand, to evaluate the performance of estimator (4), we consider the upper bound

$$\text{MSD}_{Bound} = \frac{\sigma_r^2 + \alpha^2 \sigma_w^2}{\alpha}, \tag{17}$$

derived in [8], for $a = 1$ via a distributed estimation scheme. For simplicity, we assume $\sigma_w^2 = \sigma_r^2 = \sigma^2$, and let $\beta$ in (15) be any diminishing function of $N$. Optimizing (13), (14), (15), and (17) over $\alpha$, we obtain

$$\lim_{N \to \infty} \hat{\text{MSD}}_{K_N} \approx 1.55\sigma^2 < \lim_{N \to \infty} \hat{\text{MSD}}_{S_N} = \lim_{N \to \infty} \hat{\text{MSD}}_{C_N} \approx 1.62\sigma^2 < \text{MSD}_{Bound} = 2\sigma^2,$$

which suggests a noticeable improvement in learning even in the star and cycle networks where the number of individuals and connections are in the same order.

**Regret Analysis**

We now turn to finite-time regret analysis of our methods. The average loss of all agents in predicting the state, up until time $T$, is

$$\frac{1}{T} \sum_{t=1}^{T} \frac{1}{N} \sum_{i=1}^{N} (\hat{x}_{i,t} - x_t)^2 = \frac{1}{T} \sum_{t=1}^{T} \frac{1}{N} \text{Tr}(\hat{\xi}_t \hat{\xi}_t^\mathsf{T}) \, .$$

As motivated earlier, it is not possible, in general, to drive this average loss to zero, and we need to subtract off the limit. We thus define *regret* as

$$R_T \triangleq \frac{1}{T} \sum_{t=1}^{T} \frac{1}{N} \text{Tr}(\hat{\xi}_t \hat{\xi}_t^\mathsf{T}) - \frac{1}{T} \sum_{t=1}^{T} \frac{1}{N} \text{Tr}(\hat{\Sigma}) = \frac{1}{N} \text{Tr}\left( \frac{1}{T} \sum_{t=1}^{T} \hat{\xi}_t \hat{\xi}_t^\mathsf{T} - \hat{\Sigma} \right) \, ,$$

where $\hat{\Sigma}$ is from Definition 4. We then have for the spectral norm $\| \cdot \|$ that

$$R_T \le \left\| \frac{1}{T} \sum_{t=1}^{T} \xi_t \xi_t^\mathsf{T} - \Sigma \right\|, \tag{18}$$

where we dropped the distinguishing notation between the two estimators since the analysis works for both of them. We, first, state a technical lemma from [16] that we invoke later for bounding the quantity $R_T$. For simplicity, we assume that magnitudes of both innovation and observation noise are bounded.

**Lemma 7.** *Let $\{s_t\}_{t=1}^{T}$ be an independent family of vector valued random variables, and let $H$ be a function that maps $T$ variables to a self-adjoint matrix of dimension $N$. Consider a sequence $\{A_t\}_{t=1}^{T}$ of fixed self-adjoint matrices that satisfy*

$$\left( H(\omega_1, ..., \omega_t, ..., \omega_T) - H(\omega_1, ..., \omega_t', ..., \omega_T) \right)^2 \preceq A_t^2,$$

*where $\omega_i$ and $\omega_i'$ range over all possible values of $s_i$ for each index $i$. Letting $Var = \| \sum_{t=1}^{T} A_t^2 \|$, for all $c \ge 0$, we have*

$$\mathbb{P}\left\{ \left\| H(s_1, ..., s_T) - \mathbb{E}[H(s_1, ..., s_T)] \right\| \ge c \right\} \le N e^{-c^2/8Var}.$$

**Theorem 8.** *Under conditions of Theorem 5 together with boundedness of noise $\max_{t \le T} \|s_t\| \le s$ for some $s > 0$, the regret function defined in (18) satisfies*

$$R_T \le \frac{1}{T}\left( \frac{\|\xi_0\|^2}{1 - \rho^2(Q)} \right) + \frac{1}{T}\left( \frac{2s\|\xi_0\|}{\left(1 - \rho(Q)\right)^2} \right) + \frac{1}{T}\left( \frac{s^2}{\left(1 - \rho^2(Q)\right)^2} \right) + \frac{1}{\sqrt{T}} \frac{8s^2 \sqrt{2 \log \frac{N}{\delta}}}{(1 - \rho(Q))^2}, \tag{19}$$

*with probability at least $1 - \delta$.*

We mention that results that are similar in spirit have been studied for general unbounded stationary ergodic time series in [17–19] by employing techniques from the online learning literature. On the other hand, our problem has the network structure and the specific evolution of the hidden state, not present in the above works.

# 4   The Impact of New Friendships on Social Learning

In the social learning model we proposed, agents are *cooperative* and they aim to accomplish a global objective. In this direction, the network structure contributes substantially to the learning process. In this section, we restrict our attention to estimator (5), and characterize the intuitive idea that making(losing) friendships can influence the quality of learning in the sense of decreasing(increasing) the steady state MSD of the network.

To commence, letting $\mathbf{e}_i$ denote the $i$-th unit vector in the standard basis of $\mathbb{R}^N$, we exploit the negative semi-definite, edge function matrix

$$\Delta P(i,j) \triangleq -(\mathbf{e}_i - \mathbf{e}_j)(\mathbf{e}_i - \mathbf{e}_j)^{\mathsf{T}}, \tag{20}$$

for edge addition(removal) to(from) the graph. Essentially, if there is no connection between agents $i$ and $j$,

$$P_\epsilon \triangleq P + \epsilon \Delta P(i,j), \tag{21}$$

for $\epsilon < \min\{p_{ii}, p_{jj}\}$, corresponds to a new communication matrix adding the edge $\{i,j\}$ with a weight $\epsilon$ to the network $\mathcal{G}$, and subtracting $\epsilon$ from self-reliance of agents $i$ and $j$.

**Proposition 9.** *Let $\mathcal{G}^-$ be the network resulted by removing the bidirectional edge $\{i,j\}$ with the weight $\epsilon$ from the network $\mathcal{G}$, so $P_{-\epsilon}$ and $P$ denote the communication matrices associated to $\mathcal{G}^-$ and $\mathcal{G}$, respectively. Given Assumption 1, for a fixed signal weight $\alpha$ the following relationship holds*

$$\tilde{MSD}(P,\alpha) \leq \tilde{MSD}(P_{-\epsilon},\alpha), \tag{22}$$

*as long as $P$ is positive semi-definite, and $|a| < \frac{1}{|\alpha|}$.*

Under a mild technical assumption, Proposition 9 suggests that losing connections monotonically increases the MSD, and individuals tend to maintain their friendships to obtain a lower MSD as a global objective. However, this does not elaborate on the existence of individuals with whom losing or making connections could have an immense impact on learning. We bring this concept to light in the following proposition with finding a so-called *optimal edge* which provides the most MSD reduction, in case it is added to the network graph.

**Proposition 10.** *Given Assumption 1, a positive semi-definite $P$, and $|a| < \frac{1}{|\alpha|}$, to find the optimal edge with a pre-assigned weight $\epsilon \ll 1$ to add to the network $\mathcal{G}$, we need to solve the following optimization problem*

$$\min_{\{i,j\} \notin \mathcal{E}} \sum_{k=1}^{N} \left( h_k(i,j) \triangleq \frac{z_k(i,j)\big(2(1-\alpha^2 a^2)\lambda_k(P) + 2a^2\alpha\lambda_k^2(P)\big)}{\big(1 - a^2(\lambda_k(P) - \alpha)^2\big)^2} \right), \tag{23}$$

*where*

$$z_k(i,j) \triangleq (v_k^{\mathsf{T}} \Delta P(i,j) v_k)\epsilon, \tag{24}$$

*and $\{v_k\}_{k=1}^{N}$ are the right eigenvectors of $P$. In addition, letting $\zeta_{\max} = \max_{k>1} |\lambda_k(P) - \alpha|$,*

$$\min_{\{i,j\} \notin \mathcal{E}} \sum_{k=1}^{N} h_k(i,j) \geq \min_{\{i,j\} \notin \mathcal{E}} \frac{-2\epsilon\big((1-\alpha^2 a^2)(p_{ii}+p_{jj}) + a^2\alpha([P^2]_{ii} + [P^2]_{jj} - 2[P^2]_{ij})\big)}{\big(1 - a^2\zeta_{\max}^2\big)^2}. \tag{25}$$

*Proof.* Representing the first order approximation of $\lambda_k(P_\epsilon)$ using definition of $z_k(i,j)$ in (24), we have $\lambda_k(P_\epsilon) \approx \lambda_k(P) + z_k(i,j)$ for $\epsilon \ll 1$. Based on Theorem 5, we now derive

$$\tilde{MSD}(P_\epsilon, \alpha) - \tilde{MSD}(P,\alpha) \propto \sum_{k=1}^{N} \frac{\big(\lambda_k(P_\epsilon) - \lambda_k(P)\big)\big((1-\alpha^2 a^2)(\lambda_k(P_\epsilon) + \lambda_k(P)) + 2a^2\alpha\lambda_k(P)\lambda_k(P_\epsilon)\big)}{\big(1 - a^2(\lambda_k(P) - \alpha)^2\big)\big(1 - a^2(\lambda_k(P_\epsilon) - \alpha)^2\big)}$$

$$\approx \sum_{k=1}^{N} \frac{z_k(i,j)\big(2(1-\alpha^2 a^2)\lambda_k(P) + 2a^2\alpha\lambda_k^2(P) + (1 - \alpha^2 a^2 + 2a^2\alpha\lambda_k(P))z_k(i,j)\big)}{\big(1 - a^2(\lambda_k(P) - \alpha)^2\big)\big(1 - a^2(\lambda_k(P) - \alpha + z_k(i,j))^2\big)}$$

$$= \sum_{k=1}^{N} \frac{z_k(i,j)\big(2(1-\alpha^2 a^2)\lambda_k(P) + 2a^2\alpha\lambda_k^2(P)\big)}{\big(1 - a^2(\lambda_k(P) - \alpha)^2\big)^2} + \mathcal{O}(\epsilon^2),$$

noting that $z_k(i,j)$ is $\mathcal{O}(\epsilon)$ from the definition (24). Minimizing $\tilde{\text{MSD}}(P_\epsilon, \alpha) - \tilde{\text{MSD}}(P, \alpha)$ is, hence, equivalent to optimization (23) when $\epsilon \ll 1$. Taking into account that $P$ is positive semi-definite, $z_k(i,j) \leq 0$ for $k \geq 2$, and $v_1 = \mathbf{1}_N/\sqrt{N}$ which implies $z_1(i,j) = 0$, we proceed to the lower bound proof using the definition of $h_k(i,j)$ and $\zeta_{\max}$ in the statement of the proposition, as follows

$$\sum_{k=1}^{N} h_k(i,j) = \sum_{k=2}^{N} \frac{z_k(i,j)\big(2(1-\alpha^2 a^2)\lambda_k(P) + 2a^2\alpha\lambda_k^2(P)\big)}{\big(1 - a^2(\lambda_k(P) - \alpha)^2\big)^2}$$

$$\geq \frac{1}{\big(1 - a^2\zeta_{\max}^2\big)^2} \sum_{k=2}^{N} z_k(i,j)\big(2(1-\alpha^2 a^2)\lambda_k(P) + 2a^2\alpha\lambda_k^2(P)\big).$$

Substituting $z_k(i,j)$ from (24) to above, we have

$$\sum_{k=1}^{N} h_k(i,j) \geq \frac{2\epsilon}{\big(1 - a^2\zeta_{\max}^2\big)^2} \bigg( \sum_{k=1}^{N} \big(v_k^{\mathsf{T}}\Delta P(i,j)v_k\big)\big((1-\alpha^2 a^2)\lambda_k(P) + a^2\alpha\lambda_k^2(P)\big) \bigg)$$

$$= \frac{2\epsilon}{\big(1 - a^2\zeta_{\max}^2\big)^2} \text{Tr}\bigg( \Delta P(i,j) \sum_{k=1}^{N} \big((1-\alpha^2 a^2)\lambda_k(P) + a^2\alpha\lambda_k^2(P)\big)v_k v_k^{\mathsf{T}} \bigg)$$

$$= \frac{2\epsilon}{\big(1 - a^2\zeta_{\max}^2\big)^2} \text{Tr}\bigg( \Delta P(i,j)\big((1-\alpha^2 a^2)P + a^2\alpha P^2\big) \bigg).$$

Using the facts that $\text{Tr}(\Delta P(i,j)P) = -p_{ii} - p_{jj} + 2p_{ij}$ and $\text{Tr}(\Delta P(i,j)P^2) = -[P^2]_{ii} - [P^2]_{jj} + 2[P^2]_{ij}$ according to definition of $\Delta P(i,j)$ in (20), and $p_{ij} = 0$ since we are adding a non-existent edge $\{i,j\}$, the lower bound (25) is derived. $\qquad\square$

Beside posing the optimal edge problem as an optimization, Proposition 10 also provides an upper bound for the best improvement that making a friendship brings to the network. In view of (25), forming a connection between two agents with more self-reliance and less common neighbors, minimizes the lower bound, which offers the most maneuver for MSD reduction.

## 5    Conclusion

We studied a distributed online learning problem over a social network. The goal of agents is to estimate the underlying state of the world which follows a geometric random walk. Each individual receives a noisy signal about the underlying state at each time period, so she communicates with her neighbors to recover the true state. We viewed the problem with an optimization lens where agents want to minimize a global loss function in a collaborative manner. To estimate the true state, we proposed two methodologies derived from a different decomposition of the global objective. Given the structure of the network, we provided a tight upper bound on the rate of change of the parameter which allows agents to follow the state with a bounded variance. Moreover, we computed the averaged, steady state, mean-square deviation of the estimates from the true state. The key observation was optimality of one of the estimators indicating the dependence of learning quality on the decomposition. Furthermore, defining the regret as the average of errors in the process of learning during a finite time $T$, we demonstrated that the regret function of the proposed algorithms decays with a rate $\mathcal{O}(1/\sqrt{T})$. Finally, under mild technical assumptions, we characterized the influence of network pattern on learning by observing that each connection brings a monotonic decrease in the MSD.

## Acknowledgments

We gratefully acknowledge the support of AFOSR MURI CHASE, ONR BRC Program on Decentralized, Online Optimization, NSF under grants CAREER DMS-0954737 and CCF-1116928, as well as Dean's Research Fund.

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
