[Supplementary Material]

## Supplementary Material

***Proof of Lemma 2***. We subtract (1) from (4) to get

$$\hat{x}_{i,t+1} - x_{t+1} = a\bigg(\sum_{j\in\mathcal{N}_i} p_{ij}\hat{x}_{j,t} - x_t + \alpha(y_{i,t} - \hat{x}_{i,t})\bigg) - r_t$$

$$= a\bigg(\sum_{j\in\mathcal{N}_i} p_{ij}(\hat{x}_{j,t} - x_t) + \alpha(y_{i,t} - \hat{x}_{i,t})\bigg) - r_t,$$

where we used Assumption 1 in the latter step. Replacing $y_{i,t}$ from (2) in above, and simplifying using definition of $\hat{\xi}_{i,t}$, yields

$$\hat{\xi}_{i,t+1} = a\bigg(\sum_{j\in\mathcal{N}_i} p_{ij}\hat{\xi}_{j,t} + \alpha(y_{i,t} - \hat{x}_{i,t})\bigg) - r_t$$

$$= a\bigg(\sum_{j\in\mathcal{N}_i} p_{ij}\hat{\xi}_{j,t} - \alpha\hat{\xi}_{i,t}\bigg) + (a\alpha)w_{i,t} - r_t.$$

Using definition (6) to write the above in the matrix form completes the proof for $\hat{\xi}_t$. The proof for $\tilde{\xi}_t$ follows precisely in the same fashion. $\qquad\square$

***Proof of Proposition 3***. We start by the fact that the innovation and observation noise are zero mean, so (7) implies $\mathbb{E}[\hat{\xi}_{t+1}] = Q\mathbb{E}[\hat{\xi}_t]$, and $\mathbb{E}[\tilde{\xi}_{t+1}] = Q\mathbb{E}[\tilde{\xi}_t]$. Therefore, for mean stability of the linear equations, the spectral radius of $Q$ must be less than unity. Considering the expression for $Q$ from (8), for a fixed $\alpha$ we must have

$$|a| < \frac{1}{\rho(P - \alpha I_N)} = \frac{1}{\max\{1-\alpha, |\alpha - \lambda_N(P)|\}}. \tag{26}$$

To maximize the right hand side over $\alpha$, we need to solve the min-max problem

$$\min_\alpha\bigg\{\max\{1-\alpha, |\alpha - \lambda_N(P)|\}\bigg\}.$$

Noting that $1 - \alpha$ and $\alpha - \lambda_N(P)$ are straight lines with negative and positive slopes, respectively, the minimum occurs at the intersection of the two lines. Evaluating the right hand side of (26) at the intersection point $\alpha^* = \frac{1+\lambda_N(P)}{2}$, completes the proof. $\qquad\square$

***Proof of Theorem 5***. We present the proof for $\tilde{\mathrm{MSD}}(P, \alpha)$ by observing that (7)

$$\mathbb{E}[\tilde{\xi}_{t+1}\tilde{\xi}_{t+1}^\mathsf{T}] = Q\mathbb{E}[\tilde{\xi}_t\tilde{\xi}_t^\mathsf{T}]Q^\mathsf{T} + \mathbb{E}[\tilde{s}_t\tilde{s}_t^\mathsf{T}],$$

since the innovation and observation noise are zero mean and uncorrelated. Therefore, letting $\tilde{S} = \mathbb{E}[\tilde{s}_t\tilde{s}_t^\mathsf{T}]$, since $\rho(Q) < 1$ by hypothesis, the steady state satisfies a Lyapunov equation as below

$$\tilde{\Sigma} = Q\tilde{\Sigma}Q^\mathsf{T} + \tilde{S}.$$

Let $Q = Q^\mathsf{T} = U\Lambda U^\mathsf{T}$ represent the Eigen decomposition of $Q$. Let also $u_i$ denote the $i$-th eigenvector of $Q$ corresponding to eigenvalue $\lambda_i$. Under stability of $Q$ the solution of the Lyapunov equation is as follows

$$\tilde{\Sigma} = \sum_{\tau=0}^\infty Q^\tau \tilde{S} Q^\tau$$

$$= \sum_{\tau=0}^\infty \sum_{i=1}^N \sum_{j=1}^N \lambda_i^\tau u_i u_i^\mathsf{T} \tilde{S} \lambda_j^\tau u_j u_j^\mathsf{T}$$

$$= \sum_{i=1}^N \sum_{j=1}^N u_i u_i^\mathsf{T} \tilde{S} u_j u_j^\mathsf{T} \sum_{\tau=0}^\infty \lambda_i^\tau \lambda_j^\tau$$

$$= \sum_{i=1}^N \sum_{j=1}^N \frac{u_i u_i^\mathsf{T} \tilde{S} u_j u_j^\mathsf{T}}{1 - \lambda_i \lambda_j}.$$

Therefore, the $\tilde{\text{MSD}}$ defined in 4, can be computed as

$$\tilde{\text{MSD}} = \frac{1}{N}\text{Tr}(\sum_{i=1}^{N}\sum_{j=1}^{N}\frac{u_i u_i^\mathsf{T}\tilde{S}u_j u_j^\mathsf{T}}{1-\lambda_i\lambda_j}) = \frac{1}{N}\sum_{i=1}^{N}\sum_{j=1}^{N}\frac{(u_j^\mathsf{T}u_i)(u_i^\mathsf{T}\tilde{S}u_j)}{1-\lambda_i\lambda_j} = \frac{1}{N}\sum_{i=1}^{N}\frac{u_i^\mathsf{T}\tilde{S}u_i}{1-\lambda_i^2},$$

where we used the fact that $u_j^\mathsf{T}u_i = 0$, for $i \neq j$, and $u_i^\mathsf{T}u_i = 1$, for any $i \in \mathcal{V}$. Taking into account that $Q = a(P - \alpha I_N)$ and $\tilde{S} = \sigma_r^2(\mathbf{1}_N\mathbf{1}_N^\mathsf{T}) + (a^2\alpha^2\sigma_w^2)P^2$, we derive

$$\begin{aligned}
\tilde{\text{MSD}} &= \frac{1}{N}\sum_{i=1}^{N}\frac{u_i^\mathsf{T}(\sigma_r^2(\mathbf{1}_N\mathbf{1}_N^\mathsf{T}) + (a^2\alpha^2\sigma_w^2)P^2)u_i}{1-\lambda_i^2} \\
&= \frac{1}{N}\sum_{i=1}^{N}\frac{(u_i^\mathsf{T}\mathbf{1}_N)^2\sigma_r^2}{1-\lambda_i^2} + \frac{1}{N}\sum_{i=1}^{N}\frac{a^2\alpha^2\sigma_w^2\lambda_i^2(P)}{1-\lambda_i^2} \\
&= \frac{\sigma_r^2}{1-a^2(1-\alpha)^2} + \frac{1}{N}\sum_{i=1}^{N}\frac{a^2\alpha^2\sigma_w^2\lambda_i^2(P)}{1-a^2(\lambda_i(P)-\alpha)^2},
\end{aligned}$$

where the last step is due to the facts that $\lambda_i = a(\lambda_i(P)-\alpha)$ and $\mathbf{1}_N/\sqrt{N}$ is one of the eigenvectors of $Q$ with corresponding eigenvalue $a(1-\alpha)$, so it is orthogonal to other eigenvectors, i.e., $u_i^\mathsf{T}\mathbf{1}_N = 0$, for $u_i \neq \mathbf{1}_N/\sqrt{N}$. The proof for $\tilde{\text{MSD}}$ follows in the same fashion. $\qquad\square$

***Proof of Theorem 8.*** The closed form solution of the error process (7) is,

$$\xi_{t+1} = Q^{t+1}\xi_0 + \sum_{\tau=0}^{t}Q^{t-\tau}s_\tau,$$

which implies

$$\begin{aligned}
\xi_{t+1}\xi_{t+1}^\mathsf{T} &= Q^{t+1}\xi_0\xi_0^\mathsf{T}Q^{t+1} + Q^{t+1}\xi_0\left(\sum_{\tau=0}^{t}Q^{t-\tau}s_\tau\right)^\mathsf{T} + \left(\sum_{\tau=0}^{t}Q^{t-\tau}s_\tau\right)\xi_0^\mathsf{T}Q^{t+1} \\
&\quad + \left(\sum_{\tau=0}^{t}Q^{t-\tau}s_\tau\right)\left(\sum_{\tau=0}^{t}Q^{t-\tau}s_\tau\right)^\mathsf{T},
\end{aligned} \tag{27}$$

since $Q$ is symmetric. One can see that

$$\|\frac{1}{T}\sum_{t=1}^{T}Q^t\xi_0\xi_0^\mathsf{T}Q^t\| \leq \frac{1}{T}\left(\frac{\|\xi_0\|^2}{1-\rho^2(Q)}\right),$$

and

$$\|\frac{1}{T}\sum_{t=0}^{T-1}Q^{t+1}\xi_0\left(\sum_{\tau=0}^{t}Q^{t-\tau}s_\tau\right)^\mathsf{T}\| \leq \frac{\|\xi_0\|s}{T}\sum_{t=0}^{T-1}\rho(Q)^{t+1}\sum_{\tau=0}^{t}\rho(Q)^{t-\tau} \leq \frac{1}{T}\left(\frac{s\|\xi_0\|}{(1-\rho(Q))^2}\right).$$

On the other hand, as we see in the proof of Theorem 5, letting $S = \mathbb{E}[s_\tau s_\tau^\mathsf{T}]$, we have $\Sigma = \sum_{\tau=0}^{\infty}Q^\tau SQ^\tau$. Based on definition (18), equation (27), and the bounds above, we derive

$$\begin{aligned}
R(T) &\leq \frac{1}{T}\left(\frac{\|\xi_0\|^2}{1-\rho^2(Q)}\right) + \frac{1}{T}\left(\frac{2s\|\xi_0\|}{(1-\rho(Q))^2}\right) \\
&\quad + \frac{1}{T}\left\|\sum_{t=0}^{T-1}\left(\sum_{\tau=0}^{t}Q^{t-\tau}s_\tau\right)\left(\sum_{\tau=0}^{t}Q^{t-\tau}s_\tau\right)^\mathsf{T} - \sum_{\tau=0}^{t}Q^\tau SQ^\tau\right\| + \|\frac{1}{T}\sum_{t=0}^{T-1}\sum_{\tau=t+1}^{\infty}Q^\tau SQ^\tau\|.
\end{aligned} \tag{28}$$

Let

$$H(s_0, ..., s_{T-1}) = \sum_{t=0}^{T-1}\left(\sum_{\tau=0}^{t}Q^{t-\tau}s_\tau\right)\left(\sum_{\tau=0}^{t}Q^{t-\tau}s_\tau\right)^\mathsf{T},$$

and observe that $\mathbb{E}[H(s_0,...,s_{T-1})] = \sum_{t=0}^{T-1} \sum_{\tau=0}^{t} Q^{\tau} S Q^{\tau}$. It can be verified that for any $0 \leq t < T$,

$$\|H(s_0,...,s_t...,s_{T-1}) - H(s_0,...,s'_t,...,s_{T-1})\| \leq \frac{4s^2}{\left(1 - \rho(Q)\right)^2}.$$

Thus, letting $Var = \frac{16Ts^4}{\left(1-\rho(Q)\right)^4}$, and appealing to Lemma 7, we get

$$\mathbb{P}\left\{ \left\| \sum_{t=0}^{T-1} \left( \sum_{\tau=0}^{t} Q^{t-\tau} s_\tau \right)\left( \sum_{\tau=0}^{t} Q^{t-\tau} s_\tau \right)^{\mathsf{T}} - \sum_{\tau=0}^{t} Q^{\tau} S Q^{\tau} \right\| \geq c \right\} \leq Ne^{-c^2/8Var}.$$

Setting the probability above equal to $\delta$, this implies that with probability at least $1 - \delta$, we have

$$\frac{1}{T}\left\| \sum_{t=0}^{T-1} \left( \sum_{\tau=0}^{t} Q^{t-\tau} s_\tau \right)\left( \sum_{\tau=0}^{t} Q^{t-\tau} s_\tau \right)^{\mathsf{T}} - \sum_{\tau=0}^{t} Q^{\tau} S Q^{\tau} \right\| \leq \frac{1}{\sqrt{T}} \frac{8s^2 \sqrt{2\log \frac{N}{\delta}}}{(1 - \rho(Q))^2}.$$

Moreover, we evidently have

$$\|\frac{1}{T} \sum_{t=0}^{T-1} \sum_{\tau=t+1}^{\infty} Q^{\tau} S Q^{\tau}\| \leq \frac{1}{T}\left( \frac{s^2}{\left(1 - \rho^2(Q)\right)^2} \right).$$

Plugging the two bounds above in (28) completes the proof. $\qquad\square$

***Proof of Proposition 9.*** Considering the expression for MSD in Theorem 5, we have

$$\tilde{\text{MSD}}(P, \alpha) - \tilde{\text{MSD}}(P_{-\epsilon}, \alpha) = \tilde{W}_{MSD}(P, \alpha) - \tilde{W}_{MSD}(P_{-\epsilon}, \alpha)$$

$$\propto \sum_{i=1}^{N} \frac{\left(\lambda_i(P) - \lambda_i(P_{-\epsilon})\right)\left((1 - \alpha^2 a^2)(\lambda_i(P_{-\epsilon}) + \lambda_i(P)) + 2a^2\alpha\lambda_i(P)\lambda_i(P_{-\epsilon})\right)}{\left(1 - a^2(\lambda_i(P) - \alpha)^2\right)\left(1 - a^2(\lambda_i(P_{-\epsilon}) - \alpha)^2\right)}.$$

Based on definitions (20) and (21), it follows from Weyl's eigenvalue inequality that $\lambda_k(P) - \lambda_k(P_{-\epsilon}) \leq \lambda_1\left(\epsilon\Delta P(i,j)\right) = 0$, for any $k \in \mathcal{V}$. Combined with the assumptions $P \geq 0$ and $|a\alpha| < 1$, this implies that the numerator of the expression above is always non-positive. The denominator is always positive due to stability of the error process $\tilde{\xi}_t$ in (7), and hence, $\tilde{\text{MSD}}(P, \alpha) \leq \tilde{\text{MSD}}(P_{-\epsilon}, \alpha)$. $\qquad\square$