[Reviews · NeurIPS 2013]

Submitted by Assigned_Reviewer_3

This paper addresses the problem of distributed estimation of underlying dynamic states (which is a geometric random walk) on a graph.
Estimate the underlying state is not hard is all the observations are globally known. The main challenge here is to perform local update based on information available from neighbors only.
The authors cast the problem into an online minimization of a separable, time-varying cost function and then provide two methods to perform local update on each node by employing distributed dual averaging methods. The two methods differ in the way they choose the local loss function. Then the necessary conditions for the convergence of beliefs are established.
By defining the steady state mean square deviation, the authors provide a regret analysis of the two methods and establish a bound for the regret.
This paper is technically sound.
The organization of the paper is logical and clear.
Main results of the paper seem to be correct.
As pointed out by the authors, there is a previous work focusing of distributed estimation of the underlying world given a simple random walk model [9].
[9] solves the special case where the graph is a complete graph.
This paper changes the underlying state model to a geometric random walk and the analysis are conducted on a more general graph.
However, the problem this paper target at is of limited practical impact.
The underlying state is a geometric random walk with a constant rate of change, which is assumed to be known apriori to all agents.
This constant rate of change could limit the application of the proposed methods.
It is hard to find a real world scenario where the model could be used.
Maybe the author could provide more motivating examples.
Summary: This paper is technically sound and well written. The applicable scenario of the proposed local update methods might be few.

Submitted by Assigned_Reviewer_4

The paper addresses the problem of predicting the value of an hidden dynamic variable in a network of (cooperative) agents with limited communications and noisy observations.

The communication process between neighbors limits the effect of the noise in the observations.

The authors provide a regret analysis and study the impact of new connections in the network.

To the best of my knowledge the content is original, but I am not familiar with some of the papers reported in the references.

I suggest adding some examples of practical application in the introduction.
Summary: Overall, this is an interesting paper but probably its impact in practice will be pretty limited.

Submitted by Assigned_Reviewer_5

This paper studies the problem of online learning in a social network. The state of the world is dynamic, and each individual observes a private signal about the state of the world from its connections (friends). The paper introduces two update mechanisms for estimating the true state. Steady mean-square deviation (MSD) is used to measure the difference between the estimates and the truth. The paper shows that one of the estimators recovers the optimal MSD. Furthermore, analysis on the regret and the impact of new friendship in the network is provided.

The paper is very well written and is clear.

Studying online learning in a dynamic social network is not a novel problem and is studied in [7] and [8]. The novel part is that in this paper the authors study the social network with geometric random walk and provide a learning algorithm that achieves optical MSD.

The problem is interesting. The addition to the previous work might not be very significant.
Summary: This is not my area of expertise.
The paper reads well and is clear.
The problem is interesting. Studying the specific case of geometrical random walk in comparison to previous works that study online learning in social networks with random walk might be interesting, but at the same time might be incremental.

Submitted by Assigned_Reviewer_6

This paper presents a new online learning setting: states are changing over dynamic networks. It then provides two alternative ways of updating local loss functions. Convergence and Regret analysis are given.
In general, this paper presents a novel setting of online learning. Parameters can be easily updated based on Duchi et. Al’s paper on dual averaging for distributed optimization. The idea presented in this paper looks quite impressive and the proofs seem to be correct, though the reviewer has not checked all of them. The paper is also well written.
Though as a theory paper, this paper is self-contained. It would be better to show a simulation or suggest a potential real application for this paper.
Summary: A new setting of online learning for dynamic networks is presented. The paper is self-contained, though a simulation can be provided to show the usefulness of it.
Author Feedback

Author rebuttal: We thank the reviewers for their comments. The main concern appears to be the applicability and the simplicity of the model. While we agree that the model is simple, the goal of the paper was to present a novel framework not yet considered within online learning. Because of the simple setup, we were able to find a clean solution that shows several surprising phase transitions. The first finding is the critical rate of change that characterizes whether or not the agents are able to estimate the state, even if the rate alpha is known. This critical rate depends on the network structure, highlighting the interesting distributed aspect of the problem. The second surprising fact (at least to us) was the importance of the decomposition of loss functions. Whether similar phase transitions hold for more complicated models remains to be seen.

Potential applications of the model are, for instance, in finance, where the rate of change alpha can be the known interest rate that affects the amount invested by a hedge fund. From the outside, one may only observe a noisy or corrupted version of this amount, and a network of agents aims to recover the state.

As in the paper, we remark again that the parameter alpha need not be known -- it can be learned by the agents on the fly.